# The Effect of Servant Leadership on Self-Efficacy and Innovative Behaviour: Verification of the Moderated Mediating Effect of Vocational Calling

**Yunho Ji** [ID] **and Hyun Joong Yoon** *[ID]

College of Business Administration, Kangwon National Universiy, Chuncheon 24341, Korea; yunho.ji@kangwon.ac.kr
* Correspondence: hyoon@kangwon.ac.kr

**Abstract:** This study aimed to verify the impact of servant leadership on innovative behaviour in non-governmental organisations (NGOs). It particularly investigated the role of a mediator for self-efficacy in the relationship between servant leadership and innovative behaviour. This study defined the organisational psychology-behaviour mechanism in non-profit organisations by verifying the moderated mediating effect of vocational calling in the relationship between servant leadership, self-efficacy, and innovative behaviour. The 174 pilot samples used in this study comprised community service participants in NGOs. The analysis verified the hypothesis set through causal correlations among four variables using regression analysis and the PROCESS macro developed by Hayes. Vocational calling played a moderating role in the relationship between servant leadership and self-efficacy, and vocational calling had a conditional effect on the impact of servant leadership on innovative behaviour through self-efficacy. Meanwhile, self-efficacy fully mediated servant leadership and innovative behaviour. Based on the verification of the mechanism of organisational psychology-action, this study sought ways to develop the organisation of NGOs and improve the working environment.

**Keywords:** servant leadership; vocational calling; self-efficacy; innovative behavior; NGO

## 1. Introduction

In modern society, the source of organisational competitiveness has changed from existing systems and control mechanisms to the management and utilisation of strategic human resources. By this trend, leadership has come to play a significant role in influencing, motivating and achieving organisational goals based on specialised competencies (Bartlett and Ghoshal 2002). In particular, modern organisational culture is moving away from the traditional vertical organisation model and improving company productivity through the adoption of a horizontal organisational culture that drives change and innovation by ensuring communication, autonomy, and an open working environment (Parker et al. 2001; Ramdhani et al. 2017).

Company directors play an essential role in the clear communication of company visions and goals. However, leaders also require partners or supporters who can help to create a working environment whose members are not mentally and physically exhausted (Wu et al. 2020). The organisation's continuous competitive advantage can achieve organisational democratisation by promoting universal respect and individual self-development and self-motivation. Leaders should consistently encourage members' growth and development and serve as servants to satisfy members' needs and interests. Servant leadership is an aspect of moral organisational management to develop the organisation's approach to functional tasks and participation in social relationships within a single framework (Petrovskaya and Mirakyan 2018). From this perspective, leaders' effectiveness is closely related

to qualitative factors, such as members' job satisfaction, job immersion, and self-efficacy (Van Knippenberg and Hogg 2003; Yukl 2008; Erdurmazlı 2019).

However, as individual tasks increase in diversity and complexity, vocational calling as a critical aspect of adaptation to organisations and problem solving has also increased. The sense of calling concept encompasses the pro-social value placed on contribution to others and the community rather than the pursuit of self-realisation based on individualism. In organisational psychology, a sense of calling is perceived as a predictor of positive behaviour ensuing from job satisfaction and happiness (Hall and Chandler 2005). Non-governmental organisations (NGOs)—organised by civic or private groups for the public social good—require more democratic forms of leadership and a sense of calling in their members (Lister 2003). Nevertheless, research on leadership and organisational behaviour has focused on general profit-seeking companies; there was a lack of effort to apply to non-profit organisations. In general, NGOs should apply international standards and establish an implementation system for social integration. Therefore, efforts to identify these NGOs' decision-making process mechanisms and develop their organisational behaviour are crucial to promoting their universal contribution to society. Academically, studies were mainly conducted on the improvement and development of NGO human resource management and human resource development. Still, it was not easy to find research on NGO leaders' mission, capacity, and influence. Nevertheless, the recent new paradigm shift in leadership calls for servant leadership that can contribute significantly to the organisation's integrated management and innovation in NGO (Singh 2014; Sahat et al. 2018).

It aimed to verify the predictive variables of organisational psychology and behaviour among community service participants of the Korea International Cooperation Agency (KOICA) and Good Neighbors, Korea's leading NGOs. This pilot research uses Hayes' PROCESS macro to validate moderated mediating effect of vocational calling as organisational psychology–action linked variable. Ultimately, at the humanistic psychology level, we aimed to determine the desirable qualities of NGOs' leaders and promote efficient organisational operations and strategic human resources management.

## 2. Theoretical Background

### 2.1. Servant Leadership

In leadership research, servant leadership has been established as the most human-centred model. In particular, this leadership paradigm is recognised as a key factor in the humanistic organisational role, manifesting as promoting shared values and altruistic behaviour (Sims 2018; Kumar 2018). Servant leadership denotes leaders' responsibility of care for their organisations' members by combining the words "servant" and "leader", which have contradictory meanings (Petrovskaya and Mirakyan 2018). While traditional leadership primarily expresses itself by a command/control dynamic, servant leadership is characterised by the desire to serve. Servant leadership aims to develop a social value system (Petrovskaya and Mirakyan 2018) because it shares leadership and builds trust founded on equality among organisation members.

Servant leadership is an ethical leadership distinguished from other leadership models because it places on serving people rather than treating them as tools (Erdurmazlı 2019). Servant leaders delegate their authority to demonstrate creativity and competence based on respect for the organisation's members. They also listen carefully to members' opinions and prioritise developing their organisations and their members equally (Greasley and Bocârnea 2014). Servant leadership means accepting them to empathise with the organisation's members and fully recognise their efforts and achievements even if they have to refuse (Greenleaf 1977). This attitude of empathy is beneficial for both leaders and organisational members (Bae 2009). Linda Parris and Welty Linda Parris and Peachey (2012) stressed that servant leadership is critical in non-profit organisations under its focus on service and dedication to others based on human respect. Newman et al. (2017) argued that servant leaders can foster positive sentiment within the group and improve leader-member relationships (LMX). Burton et al. (2017) suggested that servant leadership may function

as a catalyst for developing the organisation's ethical climate because members are likely to be aware of the organisation's fairness when founded on mutual trust. Poon (2006) suggests that servant leadership is fundamental to increasing mentoring and examining how it operates.

### 2.2. Self-Efficacy

Self-efficacy refers to a belief in one's ability to perform a specific task (Bandura 1989). It affects goal-seeking behaviour concerning how intensely an individual will pursue a given goal. While self-esteem constitutes respect for oneself, self-efficacy differs in that it believes in one's ability (Gardner and Pierce 1998).

Bandura (1989) emphasised the importance of social environment, human cognition, and behavioural ability for learning and development through social cognitive theory. He recognised self-efficacy as more important than self-esteem or self-satisfaction in motivating purpose-seeking behaviour. Since then, Bandura (2005) has developed into a social cognitive theory that emphasises a complementary causal model in which behavioural, cognitive, and environmental factors affect each other and create new psychological interactions. When people doubt their abilities or experience anxiety regarding their responsibilities, they may be quick to abandon or avoid complex tasks. However, individuals with high self-efficacy tend to sustain their efforts by setting higher goals and taking on more challenging or complex tasks (Feltz and Riessinger 1990). Furthermore, the causal relationship between one's sense of one's ability and role performance is motivated and organised by self-aware social and psychological conditions (Harrison et al. 1997; Dybowski et al. 2017). Motivated people are confident in themselves that they can go beyond the inverse of a particular behaviour and perform effectively in various tasks or unusual situations (Gardner and Pierce 1998). Self-efficacy affects not only the current job performance but also future organisational behavior. Therefore, self-efficacy is evaluated as a psychological variable that predicts an individual's performance in the working environment or organisational behaviour (Gist 1987).

### 2.3. Vocational Calling

The sense of calling originated as part of the Christian worldview, as a communication from God in an individual's consciousness. The calling was perceived as coming from God and legitimizing the spiritual duties assigned to the clergy within the Christian community during the Middle Ages. The meaning later expanded beyond its Christian significance to include lay professions (Seco and Lopes 2013).

The modern vocational calling is assigned social meaning by scholars in occupational psychology and organisational behaviour. It is interpreted as an altruistic desire to benefit others and society rather than pursuing one's interests (Afsar et al. 2019). In other words, the vocation concept coheres around professional values whereby the individual is grateful and satisfied and derives meaning from their work, regardless of material gain or the improvement and stability of social status. Individuals who experience a vocational calling participate in self-directed learning and innovative behaviour based on psychological ownership. In this engagement process, the individual develops the knowledge or skills required independently, to improve adaptability (Hall and Chandler 2005). Based on self-determination theory (SDT), Lee (2016) presented a sense of calling as a psychological mechanism to determine approaches to tasks and production methods. Dik et al. (2012) categorise sense of calling into three lower dimensions—transcendental calling, purpose or meaning, and pro-social orientation—that can be organised with various career variables. Hall and Chandler (2005) emphasised that individuals who experience a sense of calling adopt positive attitudes and accept and adapt to career changes with greater flexibility. Tomprou and Bankins (2019), from a positive psychology perspective, understand vocational calling as the willingness to play diverse and complex roles in and outside the working environment. Several recent studies about vocational calling have attracted attention owing to their perception of a division between the presence of calling and the

search for calling, indicating a connection between social and psychological variables (Shim and Yoo 2012).

*2.4. Innovative Behaviour*

For a company's strategic process to be reflected in its approach to decision-making, members must sympathise with and support it (Unterschuetz et al. 2008). The concept of innovative behaviour is understood as incorporating a wide range of organisational behaviours, from the creation to the implementation of ideas (Scott and Bruce 1994). Creative action should also be understood as a multi-dimensional and holistic organisational activity rather than individual creativity because it encompasses the development of ideas and the promotion, implementation, and dissemination of ideas (Janssen 2000; Rampa and Agogué 2021). In this context, innovative behaviour denotes changes in consciousness or behaviour at the individual level, such as changes in each member's duties or service methods and acquiring new skills (Li and Hsu 2016; Lee et al. 2021).

Innovation begins at the organisational level, shifts to the conscious creation of individual members, and affects job performance and behaviour changes. Therefore, innovative behaviour can be more freely expressed during stable work performance, based on intimate relationships (Qian et al. 2019). Innovative behaviour is also a form of voluntary social action that is closely related to the sense of joy and accomplishment derived from reflecting actual changes in the workplace based on the members' progressive attitudes (Kuncoro and Suriani 2018; Sameer 2018). Hughes et al. (2018) stressed that leadership—including leaders' efforts to present desirable directions and solutions, openly exchange information, and actively resolve difficulties—leads to innovative actions on the part of the organisation's members. Anderson et al. (2014) emphasised that companies can scientifically review their current organisational problems and adopt a future-oriented task design and guideline framework through the innovative actions of their members.

## 3. Research Method

*3.1. Research Models and Research Hypotheses*

This study devised a structure of interaction between variables that may meet its purpose based on the theoretical implications of servant leadership, self-efficacy, vocational calling, and innovative behaviour.

A research model was formed (illustrated in Figure 1) to analyse empirically whether servant leadership perceived by community service participants in NGO organisations affects self-efficacy and innovative behaviour. Additionally, it was considered whether vocational calling serves as a moderating mediator in the relationship between servant leadership and self-efficacy.

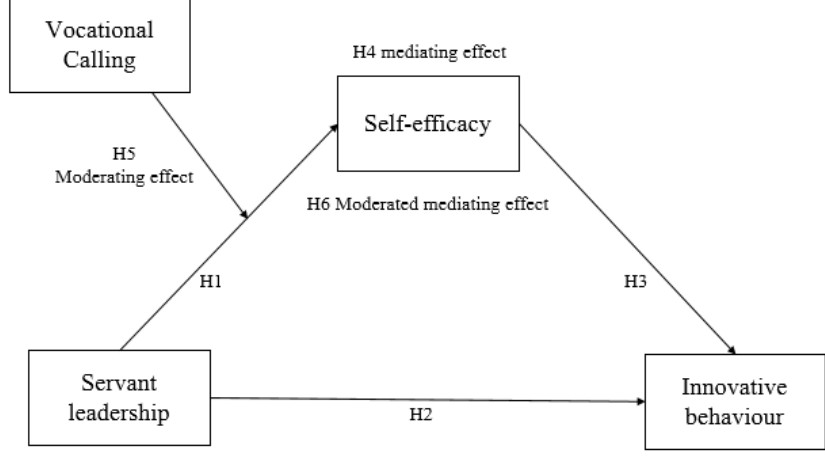

**Figure 1.** Research model.

### 3.1.1. Causality between Servant Leadership, Self-Efficacy, and Innovative Behaviour

Recent research in organisational leadership has aimed to shed light on the psychological processes through which leadership affects organisations' members and performances (Rachmawati and Lantu 2014). The theoretical paradigm that focuses on recent innovation and creative performance also analyses mutual mechanisms with psychological variables, such as leadership, self-efficacy, self-esteem, and ethical conscience. Poon (2006) explained that servant leadership provides mentees with the necessary resources to discover their talents through mentoring relationships and promotes professional knowledge and skills. Bande et al. (2016) noted that leaders' moral behaviour is emphasised, to maintain and manage adequate sales personnel to be market-oriented. Specifically, servant leadership has been proven to promote innovative action by encouraging self-reflection and self-efficacy in members and by encouraging initiative and adaptability to the market. Su et al. (2020) demonstrated that servant leadership enhances employees' internal motivation and encourages them to engage in innovative behaviour with more creative and customer-oriented services. Therefore, servant leadership can be considered as a variable that motivates intrinsic motivation and induces creative behavior.

**Hypothesis 1 (H1).** *Servant leadership will have a positive (+) effect on self-efficacy.*

**Hypothesis 2 (H2).** *Servant leadership will have a positive (+) effect on innovative behaviour.*

**Hypothesis 3 (H3).** *Self-efficacy will have a positive (+) effect on innovative behaviour.*

### 3.1.2. The Mediating Role of Self-Efficacy

Self-efficacy is a psychological mechanism related to one's job and a predictive factor that improves one's ability to control meaningful situations by expressing itself during the performance of one's duties. In particular, self-efficacy has been validated as a significant psychological variable that relies heavily on leadership and forms self-management and social relations in organisations based on services rather than manufacturing. Employees take the lead over others in their work performance, which improves their performance to maintain outstanding performance (Gardner and Pierce 1998). Qiu et al. (2020) found that self-efficacy moderates the relationship between servant leadership and service quality among employees working at chain restaurants and five-star hotels. A higher level of awareness and self-efficacy in servant leadership is associated with higher-quality service. Gong et al. (2009) demonstrated that individual learning orientation and transformational leadership are key factors that positively impact creativity, confirming that these relationships are mediated by creative self-efficacy. Zahra et al. (2017) revealed a link between ethical leadership and innovative behaviour. Furthermore, based on social learning theory, self-efficacy has been shown to play a mediating role in the relationship between the two variables.

**Hypothesis 4 (H4).** *Self-efficacy will mediate between servant leadership and innovative behaviour.*

### 3.1.3. Moderating Effect of Vocational Calling

Sense of calling is a socio-psychological variable that seeks the source of an individual's inner joy or satisfaction with a task. It is also considered to behaviour within pro-social organisations, such as social welfare and the medical industry, in terms of the psychological identity of individuals who wish to contribute to public interest through the performance of external tasks. Lee (2016) remarked that calling is a concept of achievement that emphasises self-realisation through work and is closely related to career performance. The study demonstrated that well-known workers working at the hotel's front-line exchange information influence job satisfaction through customer interactions. Seco and Lopes (2013) found

that school teachers with a sense of calling had a positive attitude toward the performance of educational public services and identified the significant moderating and moderated mediating effects of vocational calling in the relationship between authentic leadership and job commitment. Afsar et al. (2019) found that hospital nurses with a high sense of calling are more optimistic regarding higher organisational immersion and organisational civic behaviour than nurses with a low sense of calling. Park et al. (2016) proved to an insurance company's salespeople that professional self-efficacy mediates calling, job performance, and organisational civic behaviour. A sense of calling plays a moderating role in the relationship between calling and work performance-related variables.

**Hypothesis 5 (H5).** *Vocational calling will play a moderating role in the relationship between servant leadership and self-efficacy.*

**Hypothesis 6 (H6).** *Vocational calling will moderate the mediating effect of self-efficacy in the relationship between servant leadership and innovative behaviour.*

### 3.2. Measures

Servant leadership, an independent variable in this study, drew a total of five statements, including, "My boss prioritises helping me as a subordinate", referring to a relevant existing study (Spears 1995; Greasley and Bocârnea 2014; Rachmawati and Lantu 2014). The mediator, self-efficacy, drew five statements, including, "I can achieve most of the goals I set myself", referring to an earlier study (Chen et al. 2001; Cho 2016). Vocational calling, which is a moderating variable, led to five questions, including the example, "I contribute to public interest through my work", by referring to the relevant preceding study (Dik et al. 2012; Hagmaier and Abele 2012). Innovative behaviour, a dependent variable, was assessed using five items, including the example, "I devise creative ways to solve work-related problems", concerning earlier studies (Scott and Bruce 1994; Kleysen and Street 2001). These survey statements were elaborated according to this study's purposes following a preliminary review with three experts.

### 3.3. Data Collection Procedure

The survey was conducted for community service participants working in Mongolia from 1 May to 14 May 2019. Sample was collected by convenient sampling and snowball sampling, a non-probability sampling method, and was conducted by self-administration with the cooperation of the Mongolian office of the Korea International Cooperation Agency (KOICA) and Good Neighbors. A total of 200 questionnaires were collected, and 174 copies were used as final data, excluding 26 samples that were not appropriate for statistical analysis, considering the completeness, readability, and consistency of the survey. SPSS 22.0 and AMOS 22.0 statistical packages were used for empirical research. Frequency analysis was conducted to analyse the sample's demographic characteristics, and correlation analysis was conducted to assess the correlation between measurement variables before hypothesis verification. A verification factor analysis was performed to secure the measurement variable's validity and reliability analysis using the Cronbach's alpha coefficient. Finally, six hypotheses were verified by applying SPSS PROCESS Macro Models 4 and 7 devised by Hayes (2017) to achieve the study's aims.

### 4. Empirical Analysis

#### 4.1. Demographic Characteristics of Samples

Regarding the demographic characteristics of the sample (Table 1), 60 participants (34.5%) were men and 114 (65.5%) were women. For age distribution, 68 people (39.1%) were under the age of 30, 79 people (45.4%) were aged 30–40 years, 21 people (12.1) were aged 40–50 years, and six people (3.4%) were over 50. Regarding marital status, 120 (69%) were married, and 54 (31%) were unmarried. Nine (5.2%) participants were

high school graduates, 115 (66.1%) were college graduates, and 50 (28.7%) had attended graduate school. The volunteering periods involved 26 individuals (14.9%) for less than one year, 67 people (38.5%) for less than 1–3 years, 36 people (20.7%) for less than 3–5 years, 23 people (13.2%) for less than 5–7 years, and 22 people (12.6%) who worked for more than seven years. Regarding the immediate supervisors' gender, 98 were male (56.3%) and 76 female (43.7%). Regarding their nationalities, 70 (40.2%) were from Mongolia and 104 from Korea (59.8%).

**Table 1.** Demographic characteristics of samples.

| Classification | | *n* | % | Classification | | *n* | % |
|---|---|---|---|---|---|---|---|
| Gender | male | 60 | 34.5 | | Less than a year | 26 | 14.9 |
| | female | 114 | 65.5 | | 1~3 years | 67 | 38.5 |
| Age | under 30 | 68 | 39.1 | Volunteer period | 3~5 years | 36 | 20.7 |
| | 30~40 | 79 | 45.4 | | 5~7 years | 23 | 13.2 |
| | 40~50 | 21 | 12.1 | | above 7 years | 22 | 12.6 |
| | above 50 | 6 | 3.4 | | | | |
| Marital status | married | 120 | 69 | Superior's sex | male | 98 | 56.3 |
| | single | 54 | 31 | | female | 76 | 43.7 |
| Academic background | high school | 9 | 5.2 | Superior's nationality | Mongolian | 70 | 40.2 |
| | college | 115 | 66.1 | | Korean | 104 | 59.8 |
| | graduate school | 50 | 28.7 | Total | | 174 | 100 |

### 4.2. Correlation Analysis

Pearson's correlation analysis was conducted to determine the correlation between servant leadership, self-efficacy, vocational calling, and innovative behavior (Table 2). Servant leadership showed a significant correlation between vocational calling (r = 0.405, $p < 0.01$), self-efficacy (r = 0.390, $p < 0.01$), and innovative behaviour (r = 0.264, $p < 0.01$). It showed a significant correlation between self-efficacy (r = 0.684, $p < 0.01$) and innovative behaviour (r = 0.470, $p < 0.01$).

**Table 2.** Correlation analysis.

| | Servant Leadership | Vocational Calling | Self-Efficacy | Innovative Behaviour |
|---|---|---|---|---|
| Servant leadership | 1 | | | |
| Vocational calling | 0.405 ** | 1 | | |
| Self-efficacy | 0.390 ** | 0.684 ** | 1 | |
| Innovative behaviour | 0.264 ** | 0.470 ** | 0.497 ** | 1 |

** $p < 0.01$.

### 4.3. Analysis of Positive Factors and Verification of Reliability

A verification analysis was conducted to verify the validity and suitability of each variable presented in this study (Table 3). The model fit for this was judged using the significance probability of $\chi^2$, NFI, CFI, TLI, and RMSEA. The composition concept's central feasibility has been verified as having a standard value of 0.5, conceptual reliability of 0.7, and average variance extracted (AVE) of $\geq 0.5$ above standard. The suitability index for the measurement model is $\chi^2$ = 397.882 ($p < 0.001$), NFI = 0.856, IFI = 0.910, CFI = 0.909, TLI = 0.895, RMSEA = 0.091. The measurement model used in this study is generally considered to be good. The Cronbach's alpha value for all variables was deemed reliable at 0.6 or higher (Hair et al. 2014).

**Table 3.** Confirmatory factor analysis and reliability analysis of the entire composition concept.

| Latent Variable | Factor | $\lambda$ | $\alpha$ | CR | AVE |
|---|---|---|---|---|---|
| Servant leadership | prioritises members | 0.839 | | | |
| | best to help members | 0.879 | | | |
| | pays attention to the personal | 0.903 | 0.937 | 0.886 | 0.780 |
| | helps with emotional problems | 0.884 | | | |
| | cheers me up when in trouble | 0.826 | | | |
| Vocational calling | contributes to the public interest | 0.767 | | | |
| | makes the world a better place | 0.806 | | | |
| | follows the inner voice | 0.768 | 0.865 | 0.795 | 0.662 |
| | career following inner demands | 0.634 | | | |
| | fulfilling one's destiny | 0.782 | | | |
| Self-efficacy | achieves most goals | 0.766 | | | |
| | confidence in the ability to achieve | 0.808 | | | |
| | sufficient ability | 0.749 | 0.879 | 0.860 | 0.742 |
| | works better than others | 0.722 | | | |
| | can perform even in adverse situations | 0.81 | | | |
| Innovative behaviour | finds new technologies, tools, methods | 0.787 | | | |
| | uses original methods to solve problems | 0.678 | | | |
| | gains support for fundamental ideas | 0.925 | 0.913 | 0.866 | 0.753 |
| | builds empathy for innovative ideas | 0.939 | | | |
| | figures as a passionate supporter | 0.781 | | | |

chi-squared ($\chi^2$) = 397.882, normed fit index = 0.856, Tucker–Lewis index = 0.895, confirmatory factory index = 0.909, root mean square error of approximation = 0.091.

### 4.4. Hypothesis Verification

To verify whether vocational calling moderates the mediating effect of self-efficacy in the relationship between servant leadership and innovative behaviour, the PROCESS macro's Model 7 was used. A bootstrapping of 5000 was designated and the trust section was set at 95%. First, as a result of the analysis of servant leadership as an independent variable and the input of self-efficiency as a dependent variable, hypothesis 1 showed that servant leadership has a positive effect (+) on self-efficacy ($\beta$ = 0.136, $p < 0.01$). Second, as a result of analysing the impact of servant leadership on innovative behaviour, servant leadership was not associated with any significant impact on innovative behaviour ($\beta$ = 0.073, $p = 0.253$), leading to the rejection of hypothesis 2. Third, hypothesis 3 was adopted to analyse the impact of self-efficacy on innovative behaviour and demonstrated that self-efficacy has an effect of positive affection (+) on innovative behaviour ($\beta$ = 0.577, $p < 0.01$). Fourth, the interaction between servant leadership and vocational calling was significant ($\beta$ = −0.152, $p < 0.01$), and hypothesis 5—verifying the moderating effect ($R^2$ = 0.056, $p < 0.01$) was adopted (Table 4).

Fourth, servant leadership perceived by community service participants verified the mediating effect of self-efficacy in the relationship with innovative behaviour (Table 5). The total effect of the pathway between servant leadership and innovative behaviour was $\beta$ = 0.237 ($p < 0.001$), and the direct effect was $\beta$ = 0.074 ($p = 0.253$). Verification of the indirect effect of self-efficiency as a mediator using bootstrapping indicated that the indirect effect is verified because there is no zero between the bootstrap's upper and lower limits. The hypothesis that self-efficacy will play a mediating role in the relationship between servant leadership and innovative behaviour was adopted.

**Table 4.** Causal relationship between the concept of composition.

| Predictors | | | β | SE | t | p |
|---|---|---|---|---|---|---|
| **Mediator Model (Outcome Variable: Self-Efficacy)** | | | | | | |
| Constant | | | 4.334 | 0.050 | 86.752 | 0.000 |
| Servant leadership | → | Self-efficacy | 0.136 | 0.042 | 3.243 | 0.001 |
| Vocational calling | → | Self-efficacy | 0.396 | 0.058 | 6.792 | 0.000 |
| Servant leadership × vocational calling | → | Self-efficacy | −0.152 | 0.033 | −4.576 | 0.000 |
| Increase of R2 according to interaction terms | | | $R^2$ | | F | p |
| | | | 0.056 | | 20.940 | 0.000 |
| **Predictors** | | | **β** | **SE** | **t** | **p** |
| **Dependent Variable Model (Outcome Variable: Innovative Behaviour)** | | | | | | |
| Constant | | | 1.321 | 0.385 | 3.424 | 0.001 |
| Servant Leadership | → | Innovative behaviour | 0.073 | 0.064 | 1.146 | 0.253 |
| Self-efficacy | → | Innovative behaviour | 0.577 | 0.089 | 6.485 | 0.000 |

**Table 5.** The mediating effect of self-efficacy.

| Self-Efficacy | β | SE | LLCI [1] | ULCI [2] |
|---|---|---|---|---|
| Total effect | 0.237 | 0.066 | 0.106 | 0.367 |
| Direct effect | 0.074 | 0.0064 | −0.053 | 0.201 |
| Indirect effect | 0.163 | 0.052 | 0.069 | 0.273 |

[1] *LLCI* = The lower limit in the 95% confidence section of the boot indirect effect; [2] *ULCI* = Upper limit within 95% confidence section of boot indirect effect.

The conditional effect of servant leadership according to vocational calling was significant in vocational calling values from M−1SD (−1.049) to M (0.000) and not in M+1SD (1.049). If vocational calling was high, the effect of self-efficacy on innovative behaviour was not significant (Table 6).

**Table 6.** Conditional effect of servant leadership according to vocational calling.

| Vocational Calling | β | SE | t | p | LLCI | ULCI |
|---|---|---|---|---|---|---|
| −1.049 (M−1SD) | 0.2957 | 0.0598 | 4.9492 | 0.000 | 0.1778 | 0.4137 |
| 0.000 (M) | 0.1363 | 0.042 | 3.2431 | 0.0014 | 0.0533 | 0.2192 |
| 1.049 (M+1SD) | −0.0231 | 0.0489 | −0.4736 | 0.6364 | −0.1196 | 0.0733 |

β: unstandardised coefficient, *SE*: standard error, *LLCI*, *ULCI*: bias-corrected 95% confidence interval (lower limit, upper limit), M: mean, SD: standard deviation.

The area of significance determined using the Johnson-Neyman method of illumination analysis for the entire range of moderating variables is detailed in Table 7. This method offers a means of deciding which area's moderating effect according to the moderating variable is significant. The impact of servant leadership on innovative behaviour through self-efficacy was noted in areas where vocational calling values were below 0.000. In other words, in areas where the value of vocational calling is lower than 0.000, vocational calling played a role in moderating the mediating effect of self-efficacy in the relationship between servant leadership and innovative behaviour. Since the moderating impact of vocational calling was statistically significant, the results of the moderating effect to confirm the form are visualised in Figure 2. To see the pattern of meaningful interaction, vocational callings were classified into low, medium, and high groups to examine the average change. If self-efficacy was low, the higher vocational calling group had lower innovative behaviour

than the lower group, and the lower group had higher innovative behaviour even when self-efficacy was high.

**Table 7.** Conditional effect significance area of self-efficacy according to vocational calling.

| Vocational Calling | β | SE | t | p | LLCI | ULCI |
|---|---|---|---|---|---|---|
| −2.8793 | 0.5739 | 0.1119 | 5.1267 | 0 | 0.3529 | 0.7949 |
| −2.6793 | 0.5435 | 0.1058 | 5.1373 | 0 | 0.3347 | 0.7523 |
| −2.4793 | 0.5131 | 0.0997 | 5.146 | 0 | 0.3163 | 0.7099 |
| | | | ⋮ | | | |
| −0.2793 | 0.1787 | 0.0448 | 3.9875 | 0.0001 | 0.0903 | 0.2672 |
| −0.0793 | 0.1483 | 0.0426 | 3.4795 | 0.0006 | 0.0642 | 0.2325 |
| 0.1207 | 0.1179 | 0.0414 | 2.8488 | 0.0049 | 0.0362 | 0.1997 |
| 0.3207 | 0.0875 | 0.0412 | 2.124 | 0.0351 | 0.0062 | 0.1689 |
| 0.3602 | 0.0815 | 0.0413 | 1.974 | 0.05 | 0 | 0.1631 |
| 0.5207 | 0.0572 | 0.0421 | 1.3577 | 0.1764 | −0.0259 | 0.1403 |
| 0.7207 | 0.0268 | 0.044 | 0.6085 | 0.5437 | −0.06 | 0.1136 |
| 0.9207 | −0.0036 | 0.0467 | −0.0779 | 0.938 | −0.0959 | 0.0886 |
| 1.1207 | −0.034 | 0.0502 | −0.678 | 0.4987 | −0.1331 | 0.0651 |

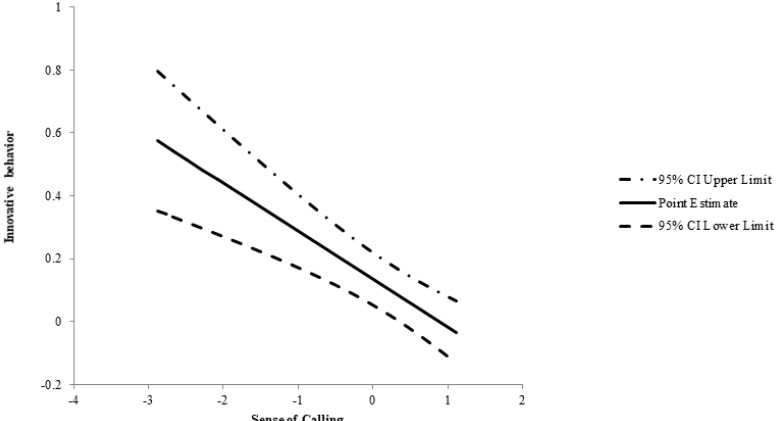

**Figure 2.** Conditional effect of servant leadership on self-efficacy at values of the moderator vocational calling.

The conditional indirect effect of vocational calling on the relationship between servant leadership and innovative behaviour was significant from M−1SD (1.049) to M (0.000) and not significant in M+1SD (1.049). A low or average vocational calling is associated with a moderated mediating effect of vocational calling on the impact of servant leadership on innovative behaviour through self-efficacy. Additionally, the moderated mediation index of vocational calling is 0.0878, and hypothesis 6, which verifies the moderated mediating effect of 95% confidence (CI) by not including zero in the lower limit and the upper limit (−0.1427, −0.0335), was supported (Table 8).

**Table 8.** Conditional effects according to vocational calling.

| Self-Efficacy | β | SE | LLCI | ULCI |
|---|---|---|---|---|
| −1.049 (M−1SD) | 0.1708 | 0.0531 | 0.0694 | 0.2766 |
| 0.000 (M) | 0.0787 | 0.0273 | 0.0279 | 0.1354 |
| 1.049 (M+1SD) | −0.0134 | 0.0195 | −0.0526 | 0.0263 |
| **Index of moderated mediation** | | SE | LLCI | ULCI |
| 0.0878 | | 0.0279 | −0.1427 | −0.0335 |

*SE*: standard error, *LLCI*, *ULCI*: bias-corrected 95% confidence interval (lower limit, upper limit).

## 5. Conclusions

This study sought ways to improve organisational development and working environments based on verifying organisational psychology behavioural mechanisms. New leadership strategies are required to strengthen the organisational capabilities of NGOs (Linda Parris and Peachey 2012). Therefore, we sought to identify the key factors necessary for evaluating NGOs' human resources and empirically analyse the conceptual composition that may be considered in developing resources. In this context, the servant leadership model was analysed regarding the relationship between self-efficacy and innovative behaviour, and the moderated mediating effect of vocational calling was verified.

First, vocational calling was found to affect the self-efficacy of members under servant leadership positively. Leaders and followers should be interdependent in situations that are not independent. Servant leadership leads members rather than managing them and was found to affect members' self-efficacy positively. Servant leadership can lead members through service and dedication, allowing them to fulfill their potential and accept responsibility without feeling burdened. Leaders can create a positive and open organisational culture only when they take the lead in gaining trust and encouraging their members to participate in challenging and demanding tasks. A leader should provide administrative support and support the necessary behaviours and capabilities for each stage of performance of his/her subordinates. A leader must also adopt an advisory approach by offering qualitative feedback rather than an arithmetical evaluation of work performance to develop confidence and self-efficacy in implementing subsequent tasks. Servant leadership is considered a significant predictor of organisational performance because non-governmental organisations are highly dependent on human resources and focus on volunteering for others.

Self-efficacy has also been shown to play a fully mediating role in the relationship between servant leadership and innovative behaviour. It means that a member's self-efficacy is a primary psychological mechanism in accepting change and innovative behaviour. Servant leaders should respect their employees' dignity and gently point out their mistakes in a manner that is not biased towards their feelings. Leaders can also encourage members to engage in lively and creative job activities by sharing their successful experiences and professional knowledge. If the leader is polite to the members and continues to mentor them in their constructive development, members will experience a desire for fulfillment and self-realisation. Companies must adopt holistic systems and support strategies to ensure that these mentor-mentee relationships are consistently maintained. Therefore, it is meaningful to verify the statistical mediating effect of self-efficacy in that team members try on their own for self-development with the support of a servant leader.

Vocational calling also plays a moderating role in the relationship between servant leadership and self-efficacy and has a conditional effect on servant leadership and innovative behaviour. The vocational calling of community service participants will serve as a source of judgment that allows them to make swift and appropriate decisions when faced with serious ethical dilemmas. However, this study's empirical findings reveal that excessive self-consciousness or sense of calling as a religious belief can hinder innovative behaviour with self-efficacy. This result is different from previous studies (Lee 2016; Afsar et al. 2019) in that sense of calling would positively affect organisational behaviour or work performance. Various community service and relief activities in NGOs require individual moral reflection and ethical awareness and interaction with bosses and colleagues and organisational dedication. Therefore, NGOs should provide career-focused education that allows community service participants to develop emotional skills, such as the sense of calling and finding value and meaning in their work and life. Additionally, the leader's counseling intervention will help, depending on the situation, or provide opportunities for formal and informal interactions and positive emotional experiences within an open cultural environment.

Based on this study's findings, recommendations for future research are presented as follows. First, the study is limited in terms of its generalisability to community service

participants dispatched to Mongolia, indicating the need for global expansion. Second, since this study is a pilot study using a small sample, additional samples need to be obtained and model verification of the structural relationship between variables. Third, for future research, follow-up studies are recommended to analyse differences between groups according to the careers, majors, and work patterns of those who have experience in community service or by introducing other forms of leadership or organisational culture a predictor of organisational psychology.

**Author Contributions:** Conceptualization, Y.J.; Data curation, H.J.Y.; Supervision, H.J.Y.; Writing—original draft, Y.J. All authors have read and agreed to the published version of the manuscript.

**Funding:** This research received no external funding.

**Institutional Review Board Statement:** Not applicable.

**Informed Consent Statement:** Not applicable.

**Data Availability Statement:** Not applicable.

**Conflicts of Interest:** The authors declare no conflict of interest.

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
