# Peer review of "The Effect of Servant Leadership on Self-Efficacy and Innovative Behaviour: Verification of the Moderated Mediating Effect of Vocational Calling"

_admsci, doi:10.3390/admsci11020039_

Round 1
Reviewer 1 Report
-The introduction. the authors need to differentiate the servant leadership concept from the other suitable styles and in details at in the literature.Why did you choose such a leadership type
-literature section need to be improved . While establishing the hypotheses, the authors must give an extensive background .It needs a comprehensive review to justify the proposed model.
-The authors has to show how they solved the common method variance problem. Because, this issue has to be handled.
-Methodology
the sampling process is not well described. Please explain exactly where were exactly the respondents? When the data were collected?how did you persuade management to conduct your research? In how many times the authors have visited such places? How did the authors contact the respondents? Explain how the survey was distributed? What is the response rate? Does the authors sample represent the population? Make sure whether the authors sample can represent the population. the authors need to justify.
-Discussions/Implications
I prefer to add A separate part for Discussion part, In particular, there is little theoretical implication discussed.
Author Response
Response to Reviewer 1
-The introduction. the authors need to differentiate the servant leadership concept from the other suitable styles and in details at in the literature. Why did you choose such a leadership type.
Response: Thank you for pointing out the issue. Strictly following your suggestions, we have modified the first sentence of the Introduction of our manuscript as below. Please see line numbers from 63 to 68 on page 3. We hope this addition adequately addresses your comments. Thank you.
"Academically, studies were mainly conducted on the improvement and development of NGO human resource management and human resource development. Still, it wasn't easy to find research on the mission, capacity and influence of NGO leaders. Nevertheless, the recent new paradigm shift in leadership calls for servant leadership that can contribute significantly to the organisation's integrated management and innovation in NGO(Singh 2014; Sahat et al. 2018).”
-literature section need to be improved . While establishing the hypotheses, the authors must give an extensive background .It needs a comprehensive review to justify the proposed model.
Response: Per your suggestion, we have modified the Literature Review and Hypotheses Development of our manuscript. Please see line number from 95 to 98, from 126 to 127 on pages 5 and 6.
“Servant leadership means accepting them in a way that empathizes with the organization's members and must fully recognize their efforts and achievements even if they have to refuse(Greenleaf 1977). This attitude of empathy is beneficial for both leaders and organisational members(Bae 2009).”
“Self-efficacy affects not only the current job performance but also future organisational behavior. Therefore, self-efficacy is evaluated as a psychological variable that predicts an individual’s performance in the working environment or organisational behaviour (Gist 1987).”
In addition, the logical basis and the argument of prior study were further supplemented for hypothesis setting on Section 3. Please see line number from 207 to 208, from 218 to 219 on pages 7 and 8. We hope these modifications adequately address your comments. Thank you.
“Therefore, servant leadership can be considered as a variable that motivates intrinsic motivation and induces creative behavior.”
“Employees take the lead over others in their work performance, which improves their performance to maintain outstanding performance (Gardner and Pierce, 1998).”
-The authors has to show how they solved the common method variance problem. Because, this issue has to be handled.
Response: There was a limit to sample collection due to the peculiarity of volunteers in NGO. Therefore we declare the research to be a pilot research with the main objective to test the method of analysis to solve this problem. In this regard, I have amended the abstract, introduction and conclusion of the study. Please see line number from 15 to 16 on page 1, from 69 to 73 on page 3, and from 438 to 440 on page 21. We hope these modifications adequately address your comments. Thank you.
[Abstract]
“The 174 pilot samples used in this study comprised community service participants in NGOs.”
[Introduction]
“It aimed to verify the predictive variables of organisational psychology and behaviour among community service participants of the Korea International Cooperation Agency (KOICA) and Good Neighbours, Korea's leading NGOs. This pilot research uses Hayes' PROCESS macro to validate moderated mediating effect of vocational calling as organisational psychology–action linked variable. ”
[Conclusion]
“ Second, since this study is a pilot study using a small sample, additional samples need to be obtained and model verification of the structural relationship between variables.”
-Methodology
the sampling process is not well described. Please explain exactly where were exactly the respondents? When the data were collected?how did you persuade management to conduct your research? In how many times the authors have visited such places? How did the authors contact the respondents? Explain how the survey was distributed? What is the response rate? Does the authors sample represent the population? Make sure whether the authors sample can represent the population. the authors need to justify.
Response: Thank you for pointing out the issue. Strictly following your suggestions, we modified the sentence as below in 3.2 Data collection procedure of our manuscript as follows. Please see line numbers from 266 to 272 on page 10 in the revised manuscript. We hope this modification adequately addresses your comments. Thank you.
"The survey was conducted for community service participants working in Mongolia from May 1 to May 14, 2019. Sample was collected by convenient sampling and snowball sampling, a non-probability sampling method, and was conducted by self-administration with the cooperation of the Mongolian office of the Korea International Cooperation Agency (KOICA) and Good Neighbours. And A total of 200 questionnaires were collected, and 174 copies were used as final data, excluding 26 samples that were not appropriate for statistical analysis, considering the completeness, readability, and consistency of the survey."
-Discussions/Implications
I prefer to add A separate part for Discussion part, In particular, there is little theoretical implication discussed.
Response: Per your suggestion, we have added the following sentences in the Conclusions of our manuscript. Please see line numbers from 405 to 407, from 417 to 419, and from 424 to 427 on pages 19 and 20, respectively. We hope these additions adequately address your suggestions. Thank you.
“Servant leadership is considered a significant predictor of organizational performance because non-governmental organizations are highly dependent on human resources and focus on volunteering for others.”
“Therefore, it is meaningful to verify the statistical mediating effect of self-efficacy in that team members try on their own for self-development with the support of a servant leader.”
“ However, this study’s empirical findings reveal that excessive self-consciousness or sense of calling as a religious belief can hinder innovative behaviour with self-efficacy. This result is different from previous studies (Lee 2016; Afsar et al. 2019) that sense of calling would positively affect organisational behaviour or work performance. ”
Response: Thank you for your valuable comments, which definitely helped us improve our manuscript quality!
Reviewer 2 Report
This article does a great job with showing each step of the mediated regression analysis and overall is well written. There are minor formatting edits needed as indicated below.
Edits are required on the references as they do not use a consistent format. For example:
- Yukl reference on line 393 is in all caps and should not be.
- some references throughout are left aligned whereas others use the justify feature
- the ampersand should be used on the reference list before the final author for all references with 2 or more authors
- Journal article titles are inconsistently capitalized
- Titles of journal names are inconsistently capitalized
Author Response
Response to Reviewer 2
This article does a great job with showing each step of the mediated regression analysis and overall is well written. There are minor formatting edits needed as indicated below.
Edits are required on the references as they do not use a consistent format. For example:
Yukl reference on line 393 is in all caps and should not be.
some references throughout are left aligned whereas others use the justify feature
the ampersand should be used on the reference list before the final author for all references with 2 or more authors
Journal article titles are inconsistently capitalized
Titles of journal names are inconsistently capitalized
Response: Thank you for recognizing the merits of our manuscript. Following your suggestions, we have modified, added, and reworked the sentences in our manuscript. Thank you.
Thank you for your valuable comments, which definitely helped us improve our manuscript quality!
Reviewer 3 Report
Remarks
While the theoretical overview and the key terms definitions are done at higher scientiffic level and the langauage of the article in general is clear and understandable, there is a big flow with the method of the research.
As it is right now the sample of 174 respondents could generalize to a maximum population of about 320 people, which is hardly proves any of the stated conclusions. And this is if we only consider random sampling, not stratified or cluster sampling. Also it is obvious that the current sample is not balanced enough to have general representativity.
A corresponding flow is the lack of definition for the general population (also for the respondents in the sample), which makes it even harder to understand the scope of generalization of the conclusions of the paper (as of now - we could only generalize to maximum of 320 population).
Another potential flow of the paper is the argumentation of the research model. As it is (presented in Figure 1) currently, it lacks enough argumentation for the general case and may be prone to hypotheses adjusting to the collected data.
Recommendations
The statistical analysis after the research model and the sample defenition is sound and done correctly so I do not have doubts that the authors would be able to upgrade the scientific quality of the paper.
There are at least two possible approaches to do so (and of course authors have their own disscretion on deciding how to proceed):
- fix the sampling approach by expand the sample size and balancing the sample while defining the studied general population, so that there is significant representativeness (probably a good idea is to aim at national level of generalization);
- declare the research to be a pilot research with main objective to test the method of research - in this case the conclusions have to be made towards the method, but not towards universal generalization of the results.
Author Response
Response to Reviewer 3
Remarks
While the theoretical overview and the key terms definitions are done at higher scientiffic level and the langauage of the article in general is clear and understandable, there is a big flow with the method of the research. As it is right now the sample of 174 respondents could generalize to a maximum population of about 320 people, which is hardly proves any of the stated conclusions. And this is if we only consider random sampling, not stratified or cluster sampling. Also it is obvious that the current sample is not balanced enough to have general representativity. A corresponding flow is the lack of definition for the general population (also for the respondents in the sample), which makes it even harder to understand the scope of generalization of the conclusions of the paper (as of now - we could only generalize to maximum of 320 population). Another potential flow of the paper is the argumentation of the research model. As it is (presented in Figure 1) currently, it lacks enough argumentation for the general case and may be prone to hypotheses adjusting to the collected data.
Recommendations
The statistical analysis after the research model and the sample defenition is sound and done correctly so I do not have doubts that the authors would be able to upgrade the scientific quality of the paper. There are at least two possible approaches to do so (and of course authors have their own disscretion on deciding how to proceed):
- fix the sampling approach by expand the sample size and balancing the sample while defining the studied general population, so that there is significant representativeness (probably a good idea is to aim at national level of generalization);
- declare the research to be a pilot research with main objective to test the method of research - in this case the conclusions have to be made towards the method, but not towards universal generalization of the results.
Response: Thank you for pointing out the issue. To address your concern, we tried to respond actively to your request and suggestions. Per your suggestion, we have added the following sentences in the Abstract, Introduction, Conclusion of our manuscript. Please see line numbers from 383 to 385 and from 396 to 398 on pages 19 and 20, respectively. We hope these additions adequately address your suggestions. Thank you.
Please see line number from 15 to 16 on page 1, from 69 to 73 on page 3 and from 438 to 440 on page 21. We hope these modifications adequately address your comments. Thank you.
[Abstract]
“The 174 pilot samples used in this study comprised community service participants in NGOs.”
[Introduction]
“It aimed to verify the predictive variables of organisational psychology and behaviour among community service participants of the Korea International Cooperation Agency (KOICA) and Good Neighbours, Korea's leading NGOs. This pilot research uses Hayes' PROCESS macro to validate moderated mediating effect of vocational calling as organisational psychology–action linked variable. ”
[Conclusion]
“ Second, since this study is a pilot study using a small sample, additional samples need to be obtained and model verification of the structural relationship between variables.”
Besides, to ensure the reliability and objectivity of the sample collection, the contents and procedure were described in more detail in the part of 3.2. Data collection procedure. Please see line numbers from 266 to 272 on page 10 in the revised manuscript. We hope this modification adequately addresses your comments. Thank you.
“The survey was conducted for community service participants working in Mongolia from May 1 to May 14, 2019. Sample was collected by convenient sampling and snowball sampling, a non-probability sampling method, and was conducted by self-administration with the cooperation of the Mongolian office of the Korea International Cooperation Agency (KOICA) and Good Neighbours. And A total of 200 questionnaires were collected, and 174 copies were used as final data, excluding 26 samples that were not appropriate for statistical analysis, considering the completeness, readable and consistency of the survey.”
Response: Per your suggestion, we have added the following sentences in the Conclusions of our manuscript. In conclusion, I described the possibility of theoretical/preliminary interpretation and appreciation through the pilot test. Please see line numbers from 405 to 407, from 417 to 419, and from 424 to 427 on pages 19 and 20, respectively. We hope these additions adequately address your suggestions. Thank you.
“Servant leadership is considered a significant predictor of organizational performance because non-governmental organizations are highly dependent on human resources and focus on volunteering for others.”
“Therefore, it is meaningful to verify the statistical mediating effect of self-efficacy in that team members try on their own for self-development with the support of a servant leader.”
“ However, this study’s empirical findings reveal that excessive self-consciousness or sense of calling as a religious belief can hinder innovative behaviour with self-efficacy. This result is different from previous studies (Lee 2016; Afsar et al. 2019) that sense of calling would positively affect organisational behaviour or work performance. ”
Thank you for your valuable comments, which definitely helped us improve our manuscript quality!
Round 2
Reviewer 3 Report
My previous remarks have been adequately worked out.